# Optimized Design of a Triangular Shear Piezoelectric Sensor Using Non-Dominated Sorting Genetic Algorithm-II(NSGA-II)

**DOI:** 10.3390/s25030803

**Published:** 2025-01-29

**Authors:** Yannan Shi, Jikun Dai

**Affiliations:** 1School of Mechanical and Equipment Engineering, Hebei University of Engineering, Handan 056038, China; daijk262728@163.com; 2School of Physics & Electronic Information Engineering, Henan Polytechnic University, Jiaozuo 454003, China

**Keywords:** piezoelectric sensors, finite element, NSGA-II, optimized design

## Abstract

A new piezoelectric sensor with a triangular shear structure was designed to conduct the deformation monitoring of geotechnical bodies in mining airspace. Firstly, a three-dimensional sensor model was developed to analyze the impact of structural parameters on resonant frequency and voltage, utilizing both finite element and experimental methods. Secondly, the NSGA-II genetic algorithm was employed to optimize the sensor’s structural parameters, focusing on resonant frequency and voltage, resulting in a Pareto optimal solution set. For the first time, the optimal parameter combination was selected by minimizing the difference method (the height of the mass block was 10.6 mm, the thickness of the piezoelectric plate was 3.29 mm, the height of the piezoelectric plate was 8.1 mm, and the height of the central column was 19 mm). The optimized sensor exhibited a 4.14% increase in resonant frequency and a 9.11% increase in voltage. Finally, the prototype was fabricated, and the effectiveness and feasibility of the design were verified through experiments. The findings indicate the sensor’s promising potential for monitoring geotechnical deformation in mining airspace regions.

## 1. Introduction

Mining is bound to form a mining hollow area, and the dynamic instability of the overlying rock in the mining hollow area changes the geological structure and stress nature, which may easily cause hidden disasters on the surface and underground, thus presenting a complex situation due to the impact of ground pressure, sudden water and other dynamic disasters [1]. Therefore, it is essential to study the deformation of the geotechnical body and to understand its pattern. The deformation monitoring of a geotechnical body allows for the accurate prediction and forecasting of the deformation and stability of the overburdened rock in the mining airspace area by detecting and analyzing the tiny vibration signals generated, and it is widely used in projects involving deep buried tunnels, large hydroelectric power stations, the deep mining of mines and other projects [2]. Enhancing sensor sensitivity within the operating frequency range enables a comprehensive and accurate acquisition of minor vibration signals, significantly improving prediction and forecasting accuracy.

At present, a great deal of research has been conducted by scholars on the design and optimization of sensors. The fiber optic accelerometer designed in the study in [3] adopts an H-type hinge structure, which further improves the sensitivity of the accelerometer by optimizing the structural parameters and can be applied to microseismic monitoring, equipment vibration monitoring and other fields. The sensors studied in [4,5] have high-sensitivity characteristics but have a relatively narrow operating frequency range and are suitable for monitoring low-frequency vibration signals. Piezoelectric sensors for microseismic monitoring were proposed in [6] for signal monitoring in a uniaxial direction. The authors in [7] designed a two-dimensional accelerometer using a flexible hinge as a spring, characterized by high sensitivity, for the detection of low-frequency vibrations with a frequency measurement range of 0.8 to 10 Hz. Other studies such as Refs. [8,9,10] proposed a new structure, which solved the contradiction between sensitivity and frequency range to varying degrees.

These studies found that there are functional compromises in the sensor, requiring a careful selection of the geometry and material of the sensor assembly. A novel frequency auto-tracking system with a measurement range of ±1 g and an enhanced frequency of 1246 Hz was introduced in [11] to tackle the limited synchronization bandwidth of traditional synchronous oscillators. Study [12] added a layer of piezoelectric structure based on a sensor with a single-voltage-layer cantilever structure. By adding Li atoms, the piezoelectric properties of the piezoelectric element were enhanced, and the voltage sensitivity was 33.1 mv/g, which was higher than the voltage sensitivity of the single-layer piezoelectric structure of 26 mv/g. Study [13] presented a high-frequency accelerometer that changed the inverse relationship between intrinsic frequency and sensitivity by means of a sensitive optical system, enhancing the accelerometer’s sensitivity across a broad spectrum of operating frequencies. The authors in [14] designed a vertical superconducting accelerometer whose intrinsic frequency could be adjusted by a coil without degrading the sensitivity of the sensor. Study [15] used the electrostatic spring softening effect to improve sensor sensitivity, which increased from 492.7 Hz/g to 2277 Hz/g. Study [16] developed an MEMS vibration sensor using single-crystal LiNbO_3_ thin-film material with strong piezoelectric constant stability. Sensitivity increased from 6.1 to 10.2 pC/g in the Z direction and from 5.2 to 6 pC/g in the *X*/*Y* direction.

After completing the preliminary sensor design, optimizing the structural parameters is essential for enhanced performance. The optimized sensor design will reduce the demand for complex structures while improving performance, which has a significant advantage in production and maintenance costs. Given that sensor design involves multi-factor, multi-objective optimization, identifying an accurate and efficient optimization method is essential. For example, Refs. [17,18,19,20] optimized the sensor design through mathematical modeling, experimental design, an adaptive genetic algorithm and response surface method and achieved good results. In contrast to the above optimization methods, the NSGA-II genetic algorithm is suitable for multi-objective optimization problems, and the solution sets derived are diverse. Sensors used for the deformation monitoring of geotechnical bodies in mining airspace areas have an operating frequency range of at least 0 to 1500 Hz and sufficient sensitivity [21].

Based on the limitations of current sensor designs in terms of sensitivity and operating frequency, a new type of integral mass sensor is proposed in this paper. A finite element model of the sensor was created and simulated to generate sample data for optimizing structural parameters. The single-factor experiment and orthogonal test method are used to reduce the number of experiments. The NSGA-II genetic algorithm is used to optimize the relationship between the design variables and the objective function. The Pareto front optimal solution is selected by minimizing the difference method for the first time. Finally, the optimized sensor performance is verified.

## 2. Sensor Working Principle and Structure

### 2.1. Working Principle

The piezoelectric effect occurs when an external force alters a crystal’s polarization state, generating a charge. The piezoelectric sensor operates on the principle of the piezoelectric effect, transforming mechanical vibrations into electrical energy. Figure 1 depicts the simplified dynamic model of the piezoelectric sensor, characterized as a single-degree-of-freedom system comprising mass (*m*), damping (*c*), and a spring (*k*) [6].

The piezoelectric sensor is fixed to the surface of the vibration generator using a bottom bolt. The fastener secures the mass block, piezoelectric plate, and central column, applying prestress to maintain the sensor’s rigid connection. When the piezoelectric sensor detects acceleration changes, the inertia of the mass block and the pressure from fasteners induce tangential stresses on the piezoelectric plate, aligned with the vibration direction. This stress causes deformation in the piezoelectric element, leading to a change in electric charge and converting the acceleration load into electrical energy. The equilibrium equations for the mass block can be derived in the equivalent system [22].(1)$$m{\ddot{x}}_{0}(t)+c{\dot{x}}_{0}(t)+kx_{0}(t)=-m{\ddot{x}}_{1}(t)$$

In the equation, m is the mass of the mass block, k is the spring constant, c is the damping coefficient, $x_{1}(t)$ represents the actual displacement of the shaking table, and $x_{0}(t)$ denotes the relative displacement of the mass block to the shaker.

The differential equation for the equivalent system derived from Equation (1) is given by the following:(2)$$m\frac{d^{2}x_{0}(t)}{dt^{2}}+c\frac{dx_{0}(t)}{dt}+kx_{0}(t)=-ma$$

In the equation, a represents the acceleration load.

The Fourier transform of Equation (2) is derived as(3)$$\frac{x_{0}(j\omega )}{a}=-\frac{{\left(1/\omega_{l}\right)}^{2}}{1-{\left(\omega /\omega_{l}\right)}^{2}+2\zeta (\omega /\omega_{l})j}$$

In the equation, $\omega_{l}$ is the natural frequency of the sensor, ζ is the damping ratio of the sensor, ω is the operating frequency of the sensor, and j is the imaginary unit.

It can be calculated as follows:(4)$$\omega_{l}=\sqrt{\frac{k}{m}}$$(5)$$\zeta =\frac{c}{2\sqrt{km}}$$

Equations (3) and (4) indicate that the natural frequency, a crucial performance metric for piezoelectric sensors, is influenced by the equivalent stiffness and the mass of the mass block.

The resonance frequency expression of the sensor is as follows:(6)$$f_{n}=\frac{1}{2\pi }\sqrt{\frac{k}{m}}$$

In the equation, $f_{n}$ is the resonant frequency.

Based on Formula (3), the sensor displacement frequency response function corresponding to each basic acceleration is represented as(7)$$\left|\frac{x_{0}}{a}\right|=\frac{1}{\omega_{l}^{2}\sqrt{{\left[1-{\left(\omega /\omega_{l}\right)}^{2}\right]}^{2}+{\left(2\zeta \omega /\omega_{l}\right)}^{2}}}$$

Assuming inertial loads on the piezoelectric sensor, Equation (8) depicts the tangential stress applied to the mass block.(8)F=ma

In the equation, F represents tangential stress.

An electrical energy proportional to the tangential stress is generated on the piezoelectric element:(9)$$Q=d_{ij}F=d_{ij}ma=d_{ij}kx_{0}(t)$$

In the equation, $d_{ij}$ is the piezoelectric constant, and Q is the generated charge.

The charge sensitivity of the piezoelectric sensor can be derived from Equations (7) and (9) as follows:(10)$$S_{Q}=\left|\frac{Q}{a}\right|=\frac{d_{ij}k}{\omega_{l}^{2}\sqrt{{\left[1-{\left(\omega /\omega_{l}\right)}^{2}\right]}^{2}+{\left(2\zeta \omega /\omega_{l}\right)}^{2}}}$$

In the equation, $S_{Q}$ is the charge sensitivity.

The equivalent capacitance of a single component can be expressed as follows [12]:(11)$$C_{a}=\frac{\varepsilon_{r}\varepsilon_{o}S}{d}$$

In the equation, $\varepsilon_{r}$ is the relative dielectric constant, $\varepsilon_{o}$ is the dielectric constant of the vacuum, S is the area of the piezoelectric plate, d is the thickness of the piezoelectric plate, and $C_{a}$ is the equivalent capacitance.

According to Formulas (10) and (11), the voltage sensitivity of the piezoelectric sensor is further obtained as follows:(12)$$S_{V}=\frac{S_{Q}}{3C_{a}}$$
where $S_{V}$ is the voltage sensitivity.

When the sensor’s operating frequency surpasses one-third of its resonant frequency, sensitivity distortion occurs, with the distortion degree peaking as it approaches the resonant frequency. The sensor’s resonance frequency sets the operating frequency range, with the maximum operating frequency generally limited to one-third of the resonance frequency [23].

### 2.2. Structure of the Sensor

Figure 2 shows the geometry of the designed sensor, which is composed of a housing shell, a conductive plate, a piezoelectric sheet, a mass block, an insulating plate, a fastener, a base, and a socket core. To optimize space utilization, the mass block features an integral arc structure and is evenly distributed across three planes.

The base center column has a piezoelectric sheet in each plane, with polarization treated in the same direction. The mass block, piezoelectric plate, and conductive plate are secured to the center post by a radial force created through the compression of the insulating plate using fasteners. Figure 3 illustrates the sensor dimensioning diagram, identifying 10 parameters (P_1_–P_10_) to define the sensor geometry. Table 1 presents the initial structural parameter values along with their respective size ranges.

### 2.3. Material Selection

Piezoelectric materials and other component materials in piezoelectric sensors have a large impact on their performance, so different piezoelectric and non-piezoelectric materials need to be analyzed for their performance. Common piezoelectric materials include quartz crystal, BaTiO_3_, and lead zirconate titanate (PZT) ceramics [24,25]. Table 2 compares their performance parameters, while Table 3 presents parameters for other component materials.

Formula (9) indicates that selecting a piezoelectric material with a high piezoelectric constant enhances the electric energy output of a piezoelectric sensor. Table 2 highlights that PZT-5H exhibits superior core parameters, including the piezoelectric constant, relative dielectric constant, and elastic modulus, compared to other piezoelectric materials. Therefore, PZT-5H was used as the sensitive material for the piezoelectric sensor.

## 3. Simulation and Experimental Analysis of Sensors

Before the finite element analysis, a three-dimensional model of the sensor was established with Solidworks2022 software. The three-dimensional model was simplified to enhance mesh division quality. When the model is solved, the system automatically establishes the contact area. The results after meshing are shown in Figure 4, including 40,948 elements and 108,509 nodes.

### 3.1. Modal Analysis

Modal analysis determines the vibration characteristics of the piezoelectric sensor structure, essential for preventing resonance. The sensor’s resonant frequency is determined to ensure safe operation within the specified frequency range. Figure 5 presents the first six simulated vibration modes of the sensor.

Figure 5 illustrates that the initial mode shape deformation predominantly affects the components near the central column. The second and third mode shapes exhibit orthogonal deformation directions, primarily located at the bottom of the central pillar, with similar deformation magnitudes. When the fourth mode shape is adopted, there is no deformation inside the sensor. The fifth mode shape is deformed at the upper position of the mass block, and the sixth mode shape is deformed at the lower position of the mass block. The sensor is deformed during the first mode shape, which affects the monitoring of the signal; in this paper, the first-order mode of the sensor is taken as the resonance frequency of 6203.4 Hz.

### 3.2. Harmonic Response Analysis

Theoretically, the resonant frequency response curve should be distorted at the resonant frequency. The harmonic response analysis employs a sinusoidal signal to excite and assess the frequency response within the 0 to 10 kHz range, using an interval of 200 points, corresponding to a frequency band of 50 Hz each [26]. When the damping coefficient is less than 0.1, the natural frequency is basically the same as the resonant frequency, and the optimal damping coefficient of 0.01 should be taken as the target in the solution [27]. Figure 6 presents the results. Before reaching one-third of the sensor’s resonant frequency, the frequency response curve approximates a straight line, indicating that the sensor’s sensitivity remains undistorted, which is beneficial for the sensor’s operation. After one-third of the resonant frequency, the frequency response curve exhibits fluctuations in amplitude, with the maximum amplitude occurring at the resonant frequency (6203.4 Hz), at which point the distortion reaches its highest level. Consequently, the sensor operates stably within a frequency range of 0 to 2067.8 Hz.

### 3.3. Piezoelectric Analysis

Piezoelectric analysis belongs to the coupling between the structure and electric field. The Ansys 17.0 version used in this study requires the installation of the corresponding piezoelectric plugin, Piezoelectric and MEMS Body, to perform piezoelectric analysis. For the piezoelectric analysis of the designed sensor, a local coordinate system must be established for each piezoelectric element, designating the Y direction as the polarization axis. The section near the central pillar of the base is set as a voltage constraint, while the opposite side is set as a voltage coupling constraint. The frequency interval is selected to be linear, with an interval point of 50 Hz, and the parameter definition is set to the simplified matrix definition option. Figure 7 shows the potential distribution cloud map under an acceleration load of 1 g.

The relationship between the output voltage of the piezoelectric plate and frequency under a 1 g load is shown in Figure 8. Within the 0~2067.8 Hz range, the sensor’s voltage output remains stable. The voltage sensitivity of the sensor, as determined by Equation (12), is 159.23 mV/g.

### 3.4. Experimental Analysis

#### 3.4.1. Single-Factor Experiments

Optimizing the sensor structure can enhance performance, as indicated by the analysis of its working principle. To enhance optimization efficiency, key factors impacting the optimization target were identified through a single-factor experiment [28]. Figure 9 presents the results.

From Figure 9, it can be seen that P_3_, P_5_, P_6_ and P_10_ play a significant role in the optimization objective, so the most influential structural parameters P_3_, P_5_, P_6_ and P_10_ will be defined as the optimization factors, and the rest of the parameters will be fixed during the optimization process.

#### 3.4.2. Orthogonal Test

As this optimization needs to consider multiple factors and multiple levels, each parameter combination is tested one by one, which not only requires a large amount of tests but also occupies resources. This study employed the orthogonal test method to efficiently organize parameter combinations, maximizing the acquisition of effective information with fewer tests to ensure result representativeness [29]. The orthogonal test factor level table, shown in Table 4, is developed by optimizing the parameter limit range in conjunction with Table 1.

Table 4 indicates the selection of the L_9_ (3^4^) orthogonal test table. In the classical design, 9 groups of experiments are needed, but the accuracy of formula fitting and subsequent algorithm optimization is not high. Therefore, on this basis, 8 groups of supplementary experiments are added by changing the position of horizontal factors, and a total of 17 groups of orthogonal experiments are carried out. The data of each group of parameters are shown in Table 5.

#### 3.4.3. Analysis of Orthogonal Test Results

The range analysis is a simple and intuitive method of multivariate data analysis. The impact of each factor on the index is assessed by determining the factor’s range value, R. A higher range value R indicates a stronger impact of the factor on the index. The range analysis of the four factors is shown in Table 6.

The range analysis table indicates that the factors influencing the resonant frequency, in order of significance, are center column height, piezoelectric plate height, piezoelectric plate thickness, and mass block height. The factors influencing voltage, in order of significance, are the thickness of the piezoelectric plate, height of the piezoelectric plate, height of the mass block, and height of the central column. Thus, during optimization, it is essential to thoroughly evaluate the appropriate values for these four factors.

## 4. Optimization of Sensor Structure Parameters

### 4.1. Data Fitting

Before optimizing the structural parameters, the corresponding mathematical expressions need to be established by fitting the available data. Utilizing orthogonal experimental design data and the 1stOpt1.5 data-fitting software, the relationships between mass height, piezoelectric sheet thickness, piezoelectric sheet height, and central column height with resonant frequency and voltage were determined.(13)$$\begin{array}{c} f(p_{3},p_{5},p_{6},p_{10})=-(35812.3354777425-3871.8199933868p_{3}-\\ 9914.58857087102p_{5}+3830.77665208794p_{6}-519.750839587234p_{10}+\\ 343.450002756377p_{3}p_{5}-109.026666103722p_{3}p_{6}+95.1333349314652p_{3}p_{10}+\\ 51.9583336966132p_{5}p_{6}+234.616667262575p_{5}p_{10}+11.7433339883023p_{6}p_{10}+\\ 74.2633310566618{p_{3}}^{2}+102.182289195812{p_{5}}^{2}-174.132500322472{p_{6}}^{2}-\\ 40.4325004457562{p_{10}}^{2}) \end{array}$$(14)$$\begin{array}{c} g(p_{3},p_{5},p_{6},p_{10})=-(-119.81529709367+34.5338337803766p_{3}+\\ 56.8291661945692p_{5}-29.3848331844326p_{6}+3.73158363924419p_{10}+\\ 4.5783333343834p_{3}p_{5}-1.20600000318398p_{3}p_{6}+0.0699999876476909p_{3}p_{10}-\\ 6.0249999901833p_{5}p_{6}+0.0825000135023937p_{5}p_{10}+0.124333329643417p_{6}p_{10}-\\ 1.16533334122811{p_{3}}^{2}-0.362499982554153{p_{5}}^{2}+2.40466666228333{p_{6}}^{2}-\\ 0.137000004481815{p_{10}}^{2}) \end{array}$$

Based on the fitting relationship, the mathematical expression of the optimization model is obtained with the maximum resonant frequency and voltage as the targets under specific constraints:(15)$$\left\{\begin{array}{c} \max\left\{\begin{array}{c} f(p_{3},p_{5},p_{6},p_{10})\\ g(p_{3},p_{5},p_{6},p_{10}) \end{array}\right.\\ s.t.\left\{\begin{array}{c} 10.5\leq p_{3}\leq 11.5,x_{1}\in 0.1N\\ 2.5\leq p_{5}\leq 3.3,x_{2}\in 0.01N\\ 7\leq p_{6}\leq 9,x_{3}\in 0.1N\\ 19\leq p_{10}\leq 21,x_{4}\in 0.1N \end{array}\right. \end{array}\right.$$

### 4.2. NSGA-II Genetic Algorithms

NSGA-II is a genetic algorithm designed for multi-objective optimization, aiming to simultaneously optimize several objective functions to achieve a set of balanced non-dominated solutions [30].

#### 4.2.1. Multi-Objective Decision Making

In multi-objective optimization problems, conflicting objectives often make it challenging to achieve an optimal solution simultaneously. At present, the conventional selection method of the optimal solution is random selection in the Pareto front area, which is subject to the subjective influence of decision makers. In this paper, we used the minimization difference method to avoid the performance of a target that was too prominent, such that each target value of the solution was more balanced to achieve the relative optimal under constraint conditions. Each solution in the Pareto optimal set was assessed and ranked using the minimization difference method. The specific decision making process is as follows:(1)The Pareto optimal solution set comprises all possible solutions. Two objective functions are utilized as evaluation criteria to construct an index matrix.(16)$$S={{\left(s_{mn}\right)}_{b}}_{\times 2}\left(m=1,2,...,b;n=1,2\right)$$

(2)Each element of the index matrix is standardized for range.


(17)
$$r_{uv}=\frac{x_{uv}-\min\left(x_{v}\right)}{\max\left(x_{v}\right)-\min\left(x_{v}\right)}$$


In the equation, $x_{uv}$ is the value of row u and column v in the data, $\max\left(x_{v}\right)$ and $\min\left(x_{v}\right)$ are the maximum and minimum values in the data of column v, respectively, and $r_{uv}$ is the standardized data.

(3)Each solution is evaluated and sequenced using the minimization difference method.


(18)
$$x_{1}=\sqrt{{\left(f_{1}\left(x_{1}\right)-f_{2}\left(x_{1}\right)\right)}^{2}}$$


In the equation, $x_{1}$ is the value of the difference between the calculated targets, and $f_{1}\left(x_{1}\right)$ and $f_{2}(x_{1})$ are the two target values for $x_{1}$.

#### 4.2.2. Optimization Results

This study employed MATLAB-R2020b to implement the NSGA-II genetic algorithm with a population size of 200, 500 iterations, a crossover probability of 0.9, and a mutation probability of 0.2. The resulting Pareto optimal solution set is illustrated in Figure 10.

#### 4.2.3. Results of Decision Making

The purpose of the optimization is to make the two optimization objectives relatively large. The solutions in the top 10 groups of the Pareto optimal solution set order obtained by using the minimization difference method are shown in Table 7.

The first-order solution considers the optimal design scheme. To verify the reliability of the optimization results, the order 1 scheme is used to reconstruct the finite element model for calculation. The finite element simulation results of the optimal scheme are then compared with those of the initial scheme, as shown in Table 8. The errors between the simulated verified values of resonance frequency and voltage and the predicted values of NSGA-II are about 1.43% and 0.14%, respectively, which indicates that reliable results are produced when multi-objective optimization is carried out using NSGA-II.

The optimal program increased the resonant frequency by 4.14% and voltage by 9.11% compared to the initial program simulation results. The device operated within a frequency range of 0.1 to 2153 Hz, with a voltage sensitivity of 173.74 mV/g as determined by Equation (12).

## 5. Results and Discussion

### 5.1. Sensor Calibration Experiment

Based on the optimization results, a prototype of the sensor was made. Through the calibration experiment, the technical indicators such as the sensitivity and operating frequency of the piezoelectric sensor are obtained, and the test device for the calibration experiment is built, as shown in Figure 11. The system consists of a computer, vibration table control, signal collector, power amplifier, vibration exciter, standard sensor, sensor for calibration, and charge amplifier. During the calibration experiment, continuous signal sampling was performed with the standard and test sensors mounted back-to-back on the exciter. Shielded cables were employed to minimize noise interference during the calibration experiment.

### 5.2. Calibration Experiment Result

The exciter’s operating frequency was restricted to 0~3000 Hz due to experimental equipment limitations. The selected test vibration frequencies were 160 Hz, 500 Hz, 1000 Hz, and 2000 Hz. The resulting output curves from the test are shown in Figure 12.

In Figure 12, the red curve represents the output signal of the standard sensor, and the green curve represents the output signal of the sensor designed in this study. Due to the significant difference in sensitivity between the two sensors, but with both sensors experiencing vibration signals at the same frequency, the sensitivity output curve of the tested sensor maintains a good sinusoidal form.

Under a 1 g acceleration load, the exciter’s excitation signal scans a frequency range of 1 to 3000 Hz. Figure 13 illustrates the real-time amplitude of the frequency response curve.

The designed sensor’s frequency response curve exhibits an amplitude variation of less than 3 dB within the 1 to 3000 Hz range. The designed sensor operates reliably within the frequency range of 0.1 to 2153 Hz.

### 5.3. Microseismic Signal Testing

The vibration experiment, depicted in Figure 14, is conducted to validate the monitoring signal’s feasibility. The difference between this platform and the calibration experimental device is that a shaking table was added, and a microseismic signal was selected for experimental verification.

The vibration signal was output through the computer software, and the vibration table generated tiny vibrations after receiving the command. Figure 15 shows the vibration signal detected by the sensor.

The experimental findings confirm the viability of the designed piezoelectric sensor for detecting vibration signals.

## 6. Conclusions

In this study, we employed the NSGA-II genetic algorithm to optimize sensor design for the deformation monitoring of geotechnical bodies in mining airspace, focusing on resonance frequency and voltage as the optimization objectives. The following conclusions are drawn:(1)Based on the single-factor experiments and orthogonal test method, ANSYS 17.0 software was used to simulate the combination of parameters, which effectively reduced the number of experiments. On this basis, range analysis was performed. The study results indicate that the resonant frequency is primarily influenced by the height of the central column, followed by the height and thickness of the piezoelectric sheet and the height of the mass block. Conversely, the voltage is most affected by the thickness of the piezoelectric plate, then the height of the piezoelectric plate and the height of the mass block, and finally, the height of the central column.(2)Using a combination of the data-fitting software 1stOpt and the NSGA-II genetic algorithm, the predicted values obtained were compared with the finite element simulation values, and the target value errors were found to be about 1.43% and 0.14%, respectively, which confirms the effectiveness of the method.(3)The optimal solution of the Pareto front was selected by the minimizing difference method for the first time. Finally, when the height of mass block was 10.6 mm, the thickness of the piezoelectric plate was 3.29 mm, the height of the piezoelectric plate was 8.1 mm and the height of the center column was 19 mm, the optimized resonance frequency and voltage increased by 4.14% and 9.11%, respectively.(4)Through calibration experiments and microseismic signal tests, the designed sensor meets the requirements of the deformation monitoring of geotechnical bodies in mining hollow areas.

The combination of finite element analysis and NSGA-II can not only solve the multi-objective optimization problem but also improve the efficiency and accuracy of optimization and provide a means to explore an effective way for the optimization design of complex systems in the future. However, due to the production and assembly of parts, the performance of the sensor still has a lot of room for improvement.

## Figures and Tables

**Figure 1 sensors-25-00803-f001:**
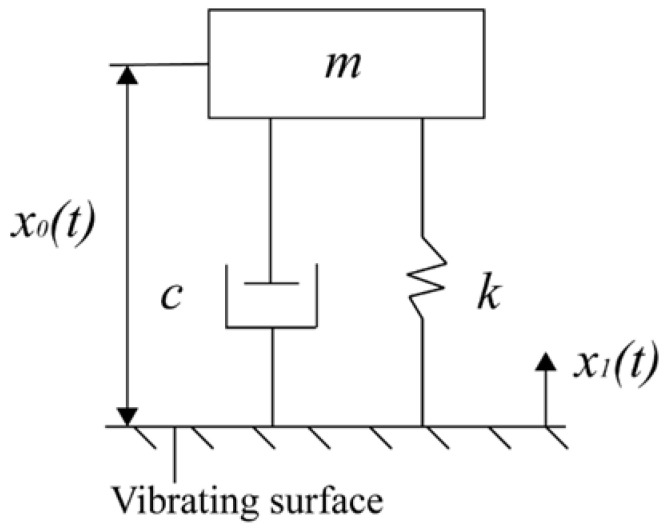
Piezoelectric sensor dynamic model.

**Figure 2 sensors-25-00803-f002:**
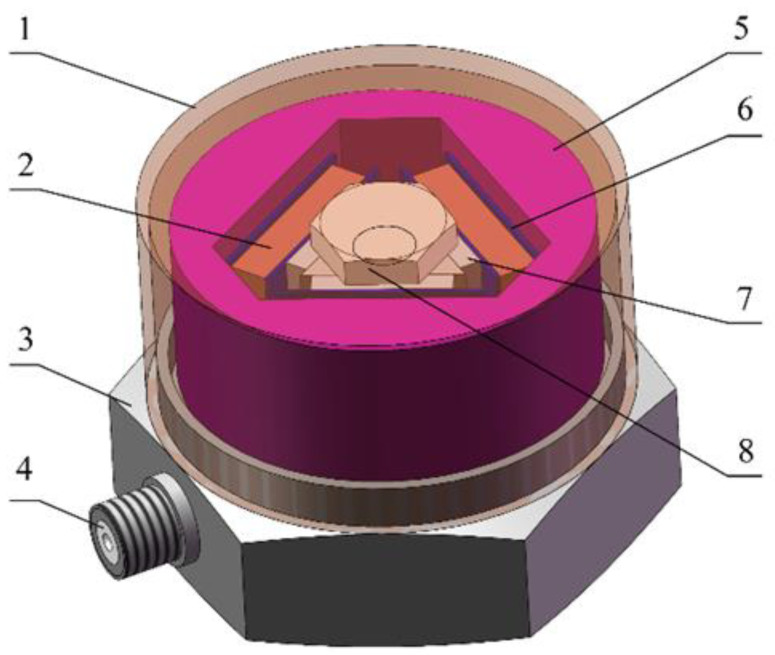
Geometric structure of the piezoelectric sensor. (1) Shell; (2) piezoelectric plate; (3) base; (4) socket core; (5) mass block; (6) conductive plate; (7) insulating plate; (8) fastener.

**Figure 3 sensors-25-00803-f003:**
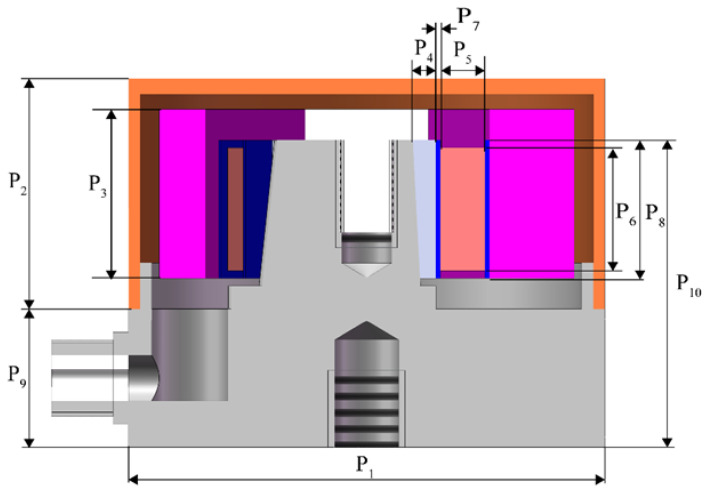
The 10 design variables of the sensor.

**Figure 4 sensors-25-00803-f004:**
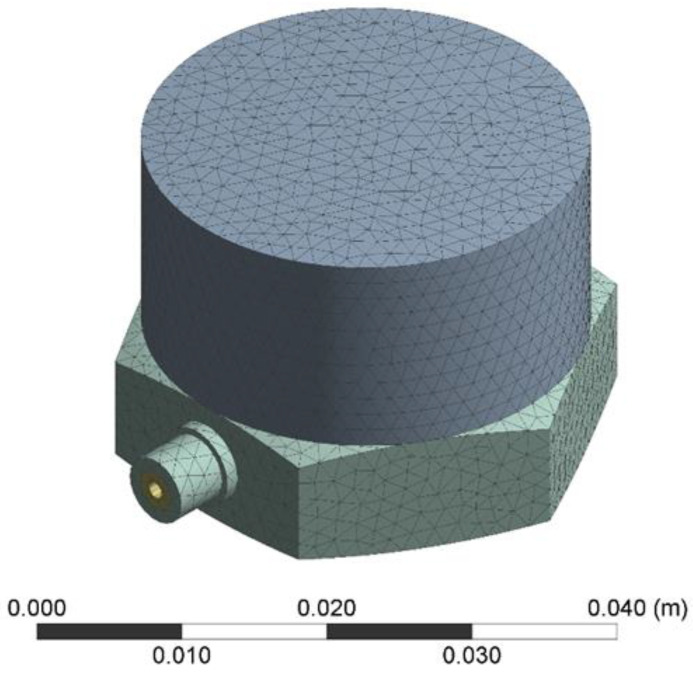
The simulation model of the sensor.

**Figure 5 sensors-25-00803-f005:**
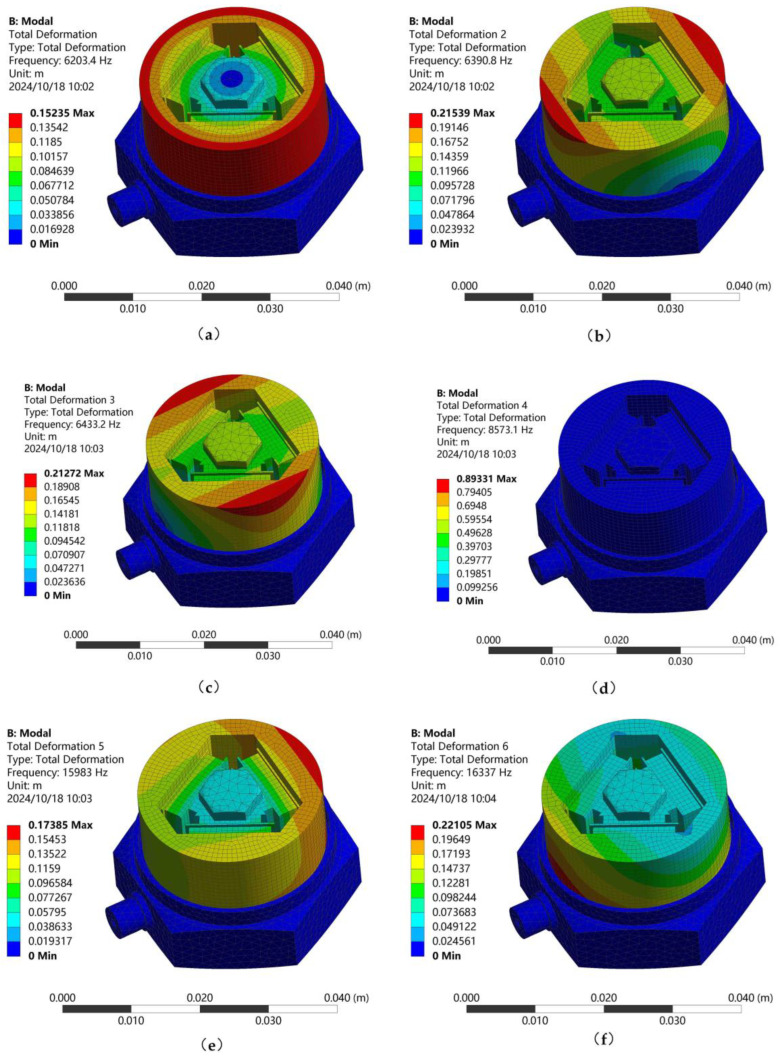
The initial six mode shapes of the sensor. (**a**) The first modal shape; (**b**) The second mode shape; (**c**) The third mode shape; (**d**) The fourth mode shape; (**e**) The fifth mode shape; (**f**) The sixth mode shape.

**Figure 6 sensors-25-00803-f006:**
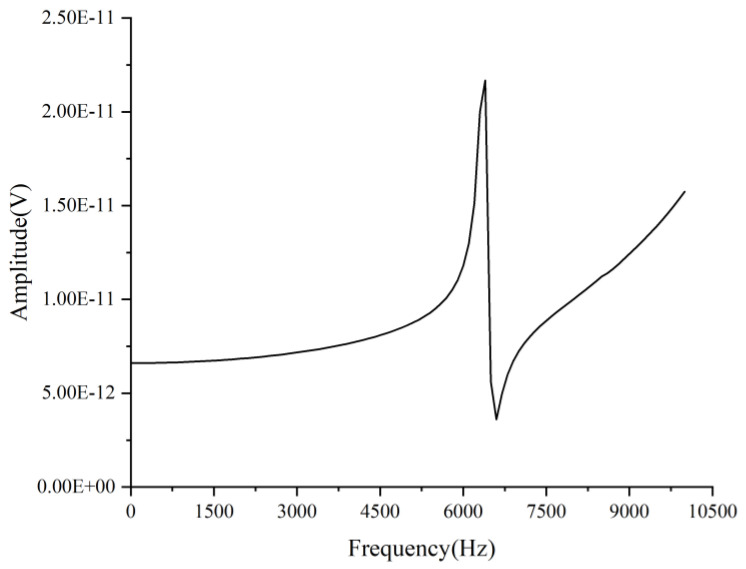
Resonant frequency response of the sensor.

**Figure 7 sensors-25-00803-f007:**
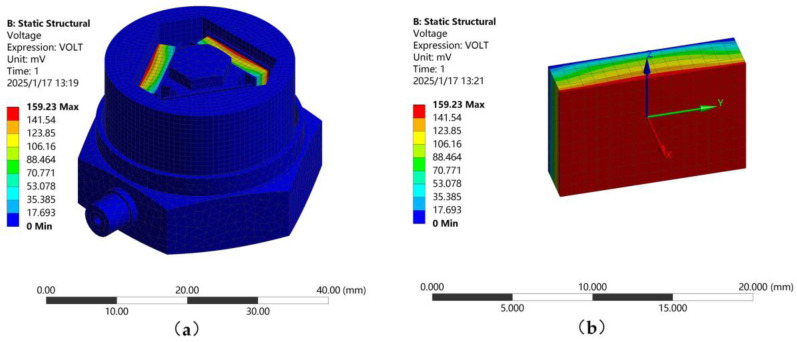
Voltage profile of the sensor. (**a**) Overall voltage profile of the sensor; (**b**) Voltage distribution of piezoelectric plate.

**Figure 8 sensors-25-00803-f008:**
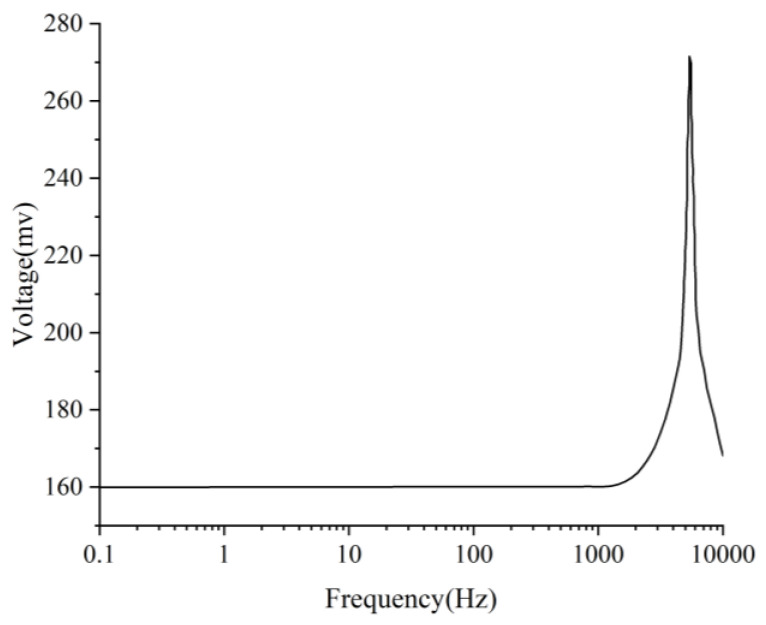
Frequency voltage diagram of piezoelectric plate.

**Figure 9 sensors-25-00803-f009:**
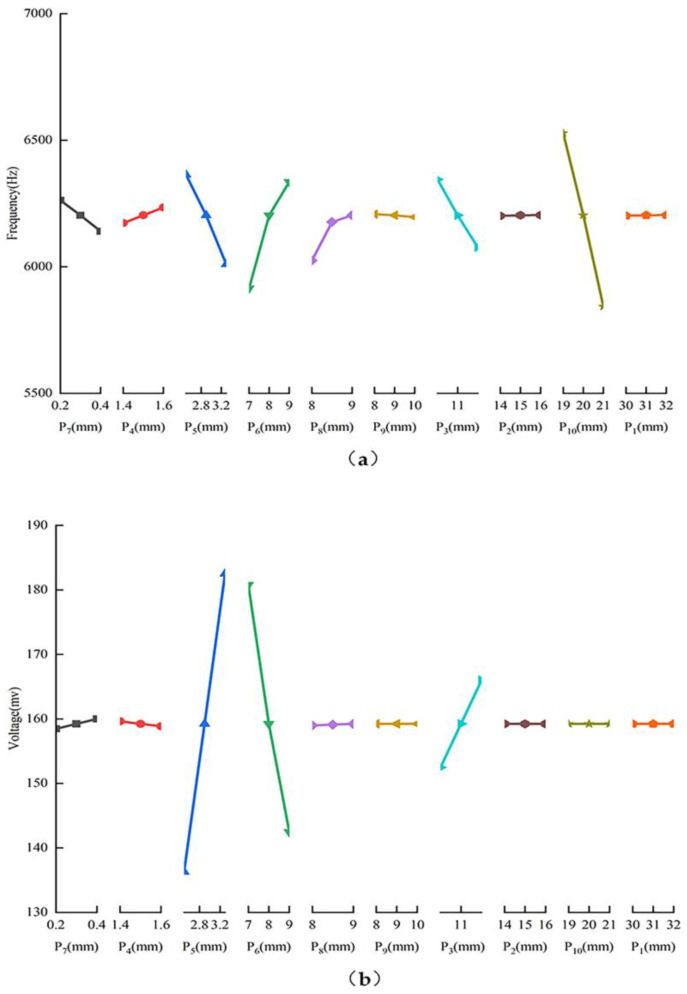
The influence of different structural parameters on optimization objectives. (**a**) The influence of each dimension parameter on the resonant frequency; (**b**) The influence of each dimension parameter on the voltage value.

**Figure 10 sensors-25-00803-f010:**
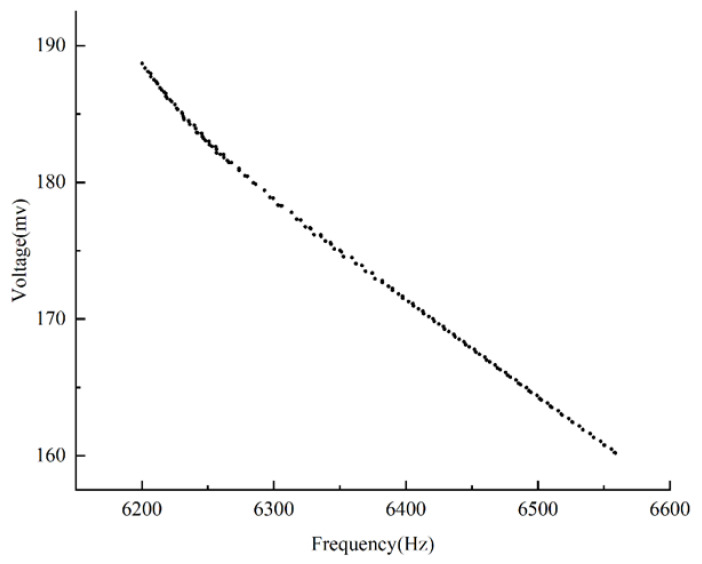
Pareto optimal solution set result.

**Figure 11 sensors-25-00803-f011:**
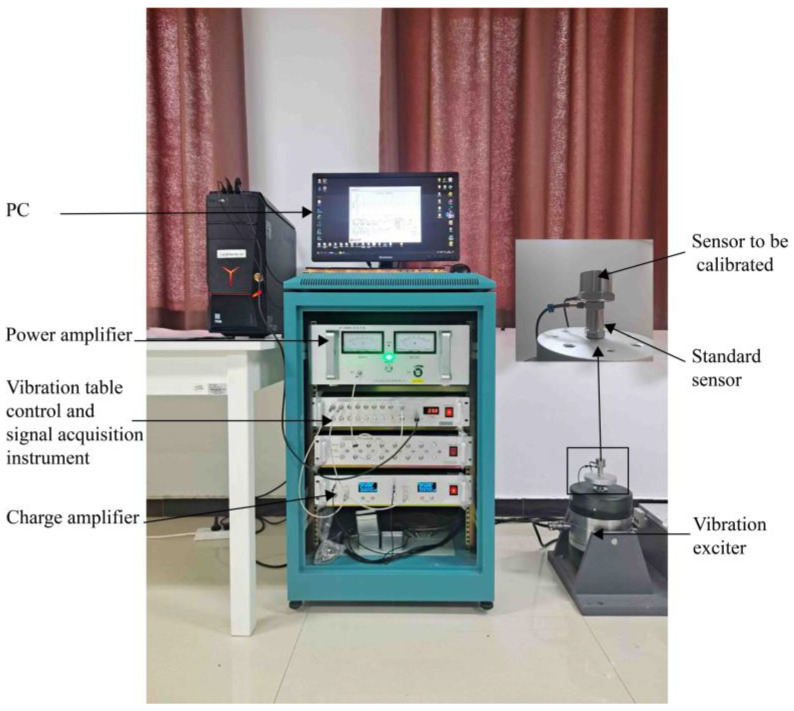
Calibration of experimental test equipment.

**Figure 12 sensors-25-00803-f012:**
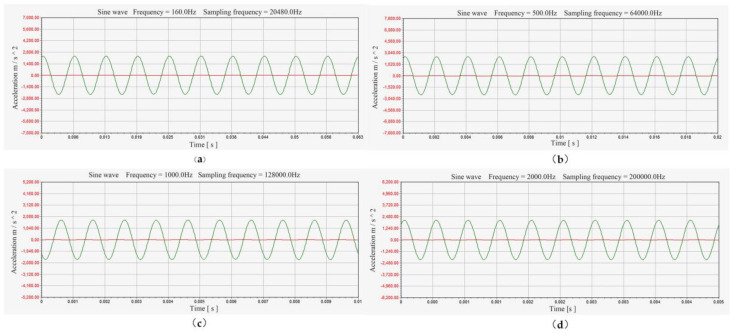
Dynamic output characteristic curve of sensor at different frequencies. (**a**) 160 Hz; (**b**) 500 Hz; (**c**) 1000 Hz; (**d**) 2000 Hz.

**Figure 13 sensors-25-00803-f013:**
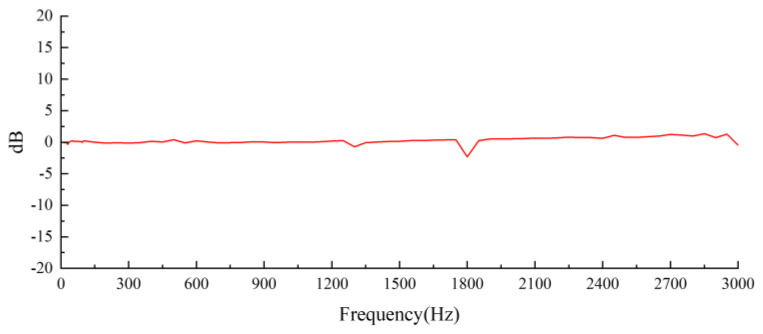
Sensor frequency response curve.

**Figure 14 sensors-25-00803-f014:**
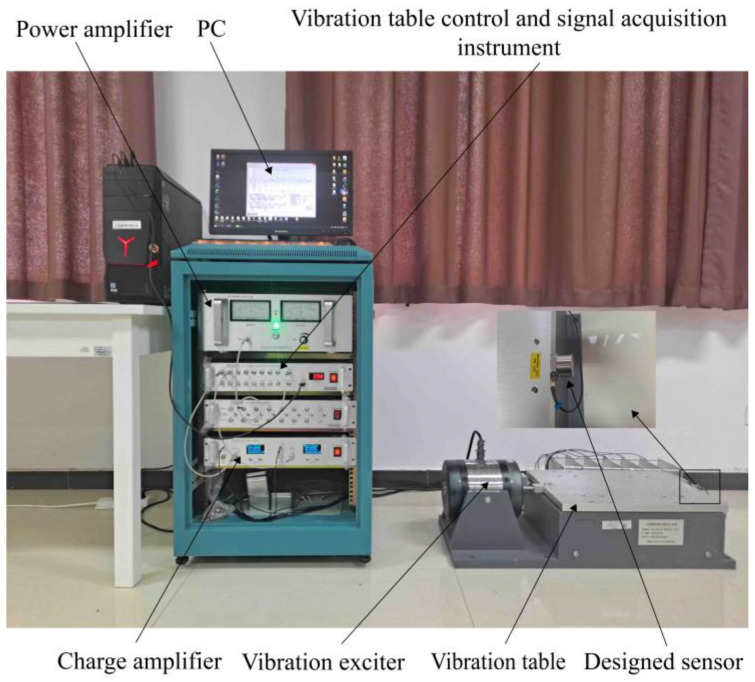
Physical diagram of sensor vibration signal test device.

**Figure 15 sensors-25-00803-f015:**
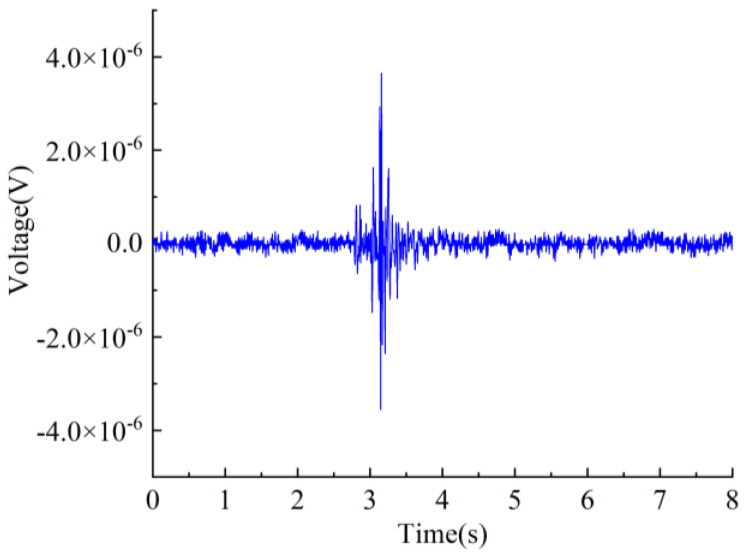
The sensor is engineered to detect the vibration signal waveform.

**Table 1 sensors-25-00803-t001:** Dimensions of various parts of the sensor.

Dimension Name	Design Parameter	Initial Value (mm)	Size Range (mm)
Outside diameter of base inner cut circle	P_1_	31	30–32
Height of shell	P_2_	15	14–15
Height of mass block	P_3_	11	10.5–11.5
Thickness of insulating plate	P_4_	1.5	1.4–4.6
Thickness of piezoelectric plate	P_5_	2.9	2.5–3.3
Height of piezoelectric plate	P_6_	8	7–9
Thickness of conductive plate	P_7_	0.3	0.2–0.4
Height of conductive plate	P_8_	9	8–9
Height of foundation	P_9_	9	8–10
Height of the central column	P_10_	20	19–21

**Table 2 sensors-25-00803-t002:** Comparison of the related parameters and piezoelectric performance.

Performance Parameter	Material				
	Quartz	BaTiO_3_	PZT-4	PZT-5H	PZT-8
Piezoelectric constant (pC/N)	$d_{11}$ = 2.31	$d_{15}$ = 260	$d_{15}$ = 410	$d_{15}$ = 741	$d_{15}$ = 410
Relative dielectric constant	4.52	1200	1050	1700	1000
Density (kg/m^3^)	2650	5500	7450	7500	7600
Modulus of elasticity (GPa)	80	110	83	117	123
Curie point (°C)	550	80	328	193	300

**Table 3 sensors-25-00803-t003:** Component materials and parameters.

Component	Materials	Young’s Modulus (Pa)	Density (kg/m^3^)	Poisson’s Ratio
Base	stainless steel	2 × 10^11^	7860	0.3
Shell	stainless steel	2 × 10^11^	7860	0.3
Conductive plate	copper alloy	1.1 × 10^11^	8300	0.34
Insulating plate	alumina ceramic	3 × 10^11^	3600	0.23
Mass block	tungsten alloy	3.4 × 10^11^	17,500	0.27
Fastener	stainless steel	2 × 10^11^	7860	0.3
Socket core	polyethylene	1.1 × 10^9^	950	0.42

**Table 4 sensors-25-00803-t004:** Orthogonal test factor level table.

Level Factors	P_3_ (mm)	P_5_ (mm)	P_6_ (mm)	P_10_ (mm)
1	10.5	2.5	7	19
2	11	2.9	8	20
3	11.5	3.3	9	21

**Table 5 sensors-25-00803-t005:** Orthogonal test table.

Serial Number	P_3_ (mm)	P_5_ (mm)	P_6_ (mm)	P_10_ (mm)	Resonant Frequency (Hz)	Voltage (mv)
1	10.5	2.5	7	19	6606.1	148.22
2	10.5	2.9	8	20	6341.5	152.48
3	10.5	2.9	9	21	6080.5	136.45
4	10.5	3.3	8	20	6144.7	174.81
5	10.5	3.3	9	21	5957.9	156.57
6	11	2.5	8	21	5976.8	136.29
7	11	2.5	9	20	6464.1	121.83
8	11	2.9	8	19	6522.9	159.23
9	11	2.9	9	19	6538	142.46
10	11	3.3	7	20	5643	207.04
11	11	3.3	7	21	5369.2	207.04
12	11.5	2.5	8	21	5849.8	142.11
13	11.5	2.5	9	20	6252.9	127.01
14	11.5	2.9	7	20	5789.9	188.52
15	11.5	2.9	7	21	5487.7	188.52
16	11.5	3.3	8	19	6184.7	190.19
17	11.5	3.3	9	19	6206.5	170.24

**Table 6 sensors-25-00803-t006:** Range analysis table.

Projects	Level	P_3_		P_5_		P_6_		P_10_	
		Resonant Frequency (Hz)	Voltage (mv)	Resonant Frequency (Hz)	Voltage (mv)	Resonant Frequency (Hz)	Voltage (mv)	Resonant Frequency (Hz)	Voltage (mv)
K	1	31,130.7	768.53	31,149.7	675.46	28,895.9	939.34	32,058.2	810.34
	2	36,514	973.89	36,760.5	967.66	37,020.4	955.11	36,636.1	971.69
	3	35,771.5	1006.59	35,506	1105.89	37,499.9	854.56	34,721.9	966.98
K avg	1	6226.14	153.71	6229.94	135.09	5779.18	187.87	6411.64	162.07
	2	6085.67	162.31	6126.75	161.28	6170.07	159.18	6106.02	161.95
	3	5961.92	167.76	5917.67	184.31	6249.98	142.43	5786.98	161.16
R		264.22	14.06	312.27	49.22	470.8	45.44	624.66	0.9
Number of levels		3		3		3		3	
Number of repeats		5		5		5		5	

**Table 7 sensors-25-00803-t007:** Decision making to obtain the top 10 solutions in order in the Pareto optimal solution set.

Ordinal Number	Optimization Variables				Objectives		Target Difference
	P_3_ (mm)	P_5_ (mm)	P_6_ (mm)	P_10_ (mm)	Resonant Frequency (Hz)	Voltage (mv)	
1	10.6	3.29	8.1	19	6369.5	173.49	0.005173
2	10.5	3.29	8	19	6366.89	173.91	0.016887
3	10.5	3.28	8	19	6374.52	173.36	0.023479
4	10.6	3.3	8.1	19	6362.27	174.05	0.033467
5	10.6	3.28	8.1	19	6376.73	172.94	0.044978
6	10.5	3.3	8	19	6359.27	174.48	0.0572554
7	10.5	3.27	8	19	6382.18	172.8	0.065572
8	10.5	3.23	7.9	19	6382.2	172.65	0.070623
9	10.7	3.28	8.1	19	6352.8	174.56	0.079058
10	10.5	3.3	8.1	19	6386.9	172.39	0.09398

**Table 8 sensors-25-00803-t008:** Results of the comparison between the optimal and initial scenarios.

Program Category	P_3_ (mm)	P_5_ (mm)	P_6_ (mm)	P_10_ (mm)	Resonant Frequency (Hz)	Voltage (mv)
Initial program	11	2.9	8	20	6203.4	159.23
Optimal program	10.6	3.29	8.1	19	6460.3	173.74

## Data Availability

The data presented in this study are available in the article.

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
