# Peer review of "Optimized Design of a Triangular Shear Piezoelectric Sensor Using Non-Dominated Sorting Genetic Algorithm-II(NSGA-II)"

_sensors, 2025, doi:10.3390/s25030803_

Round 1
Reviewer 1 Report
Comments and Suggestions for Authors
The authors propose a new piezoelectric sensor with a triangular shear structure so they showed some results with specific model data of the piezoelectric sensors. A 4.14% increase in resonant frequency and a 9.11% increase in voltage look not bad results. The previous studies about the the design and optimization of sensors are well described so I can understand that there are functional compromises in the sensor, requiring careful selection of the geometry and material of the sensor assembly. The previous studies from References [3-15] look good so I can understand the novel idea about the piezoelectric sensor with a triangular shear structure. The mathematical analysis of the optimization model and simulation results about the piezoelectric sensor with a triangular shear structure show good to be understood. There are no English grammar issues and I can fully understand the topic of the research. However, there are some minor questions for the results in the submitted manuscript. Thus, I can conclude that the submitted manuscript can be minor revision before acceptance.
1. What is the abbreviated name of NSGA-II ? The authors must define it before using that.
2. From Equations (1) to (12), the authors forgot to mention the definition of some labels. Please provide them.
3. In Figure 6, amplitude unit is wrong. It must be V.
4. Figures 7, 8, 9, 14, and 15 have very small size fonts. Please increase the font size of the labels.
5. Why the authors provide the Multi-Objective Decision Making ? It looks like unnecessary.
6. In Figure 10, are there any saturation points ? They are keeping decreasing.
7. Is there any possible for Fourier transform (FFT) result for Figure 15 ? From FFT result, the sensor frequency is supposed to be around 1800 Hz.
8. The authors had better describe the limitation of the proposed research or future work briefly.
9. How to obtain the dynamic output curve ? Is it to be calculated or measured ?
10. In Table 7, target difference is between simulated and measured results ?
Author Response
Dear Reviewer,
Thank you for your detailed review of our work and valuable suggestions. The following is our specific response:
- What is the abbreviated name of NSGA-II ? The authors must define it before using that.
Answer: Thank you for pointing this out. I agree with this comment. The abbreviation of the second generation non-dominated sorting genetic algorithm is NSGA-II. I have modified it and marked it in red in the title.
- From Equations (1) to (12), the authors forgot to mention the definition of some labels. Please provide them.
Answer: Thank you for pointing this out. I have modified Equation(1), (3), (6), (8), (9), (11), (12) and marked it in red.
- In Figure 6, amplitude unit is wrong. It must be V.
Answer: I agree with this comment. The modification has been completed. I made a unit modification in Figure 6. Due to the adjustment of the figure, the details are shown in Figure 4.
- Figures 7, 8, 9, 14, and 15 have very small size fonts. Please increase the font size of the labels.
Answer: Agree. Figures 7, 8, 9, 14 and 15 have been modified to increase the font size. Due to the adjustment of the figure, the details are shown in Figure 5,6,7,12, and 13.
- Why the authors provide the Multi-Objective Decision Making ? It looks like unnecessary.
Answer: Thank you for pointing this out. With the advancement of technology and social development, the problems in various complex systems are often no longer limited to the optimization of a single objective, but need to make trade-offs and choices among multiple optimization objectives. Multi-objective decision-making is a scientific method to find the best balance between multiple conflicting objectives. At present, the conventional selection method of the optimal solution is random selection in the Pareto frontier region, which is greatly influenced by the decision makers. The multi-objective decision-making used in this paper can make the solution of each target value more balanced and not subject to the subjective influence of decision makers, so this method is needed in this paper. At the same time, it also provides a good idea for future scientific research.
- In Figure 10, are there any saturation points ? They are keeping decreasing.
Answer: Thank you for pointing this out. The existence of the saturation point can be used as a reference solution for the final optimization, because it reflects a reasonable compromise in multi-objective optimization. However, it can be seen in Figure 10 that there is no significant inflection point in the shape of the Pareto front, and all solutions are uniformly and continuously distributed, so there is no saturation point. This situation increases the complexity of the optimization process, making it difficult for decision makers to determine a specific optimal solution. Of course, it is not unsolvable. Multi-objective decision-making can be introduced to select practical solutions from the Pareto frontier without saturation points to provide support for the final application.
- Is there any possible for Fourier transform (FFT) result for Figure 15 ? From FFT result, the sensor frequency is supposed to be around 1800 Hz.
Answer: Thank you for pointing this out. The experimental data are reliable. It can also be seen in the diagram that although the frequency response curve of the designed sensor changes at 1800 Hz, the amplitude of the change is within ± 3 dB, and the sensor can work stably.
- The authors had better describe the limitation of the proposed research or future work briefly.
Answer: Agree. The modification has been completed and marked in red font in the summary section.
- How to obtain the dynamic output curve ? Is it to be calculated or measured ?
Answer: Thank you for pointing this out. The dynamic output curve is the performance of the output signal changing with time when the sensor responds to the input excitation, which can be obtained directly by experimental measurement.
- In Table 7, target difference is between simulated and measured results ?
Answer: Thank you for pointing this out. The difference value here is the difference between the two optimization target values after range standardization. The smaller the difference value is, the more balanced the optimization target is. Of course, when one goal is too prominent and the other goal is worse, the difference value is greater, but this is not what we want to see.
Thank you again for your attention and valuable suggestions for our research, which is of great help to us to improve the quality of the paper and express the significance of the research.
Kind regards,
Jikun Dai

Reviewer 2 Report
Comments and Suggestions for Authors
The abstract is a well-structured overview of the development and optimization of a new piezoelectric sensor with a triangular shear structure for geotechnical deformation monitoring in mining airspace.
The introduction provides a thorough of the challenges and research directions. The topic is presented with clarity, supported by an extensive review of recent advancements and methodologies. Introduction effectively highlights the significance of studying the dynamic instability caused by mining activities and its implications for surface and underground safety. This sets a strong context for the research. In my opinion summarizing similar studies together could reduce redundancy and enhance readability. While the introduction underscores the technical advancements, it could elaborate more on the broader impacts of the proposed sensor design, such as its potential implications for cost-efficiency or industrial adoption. Authors should emphasize the unique aspects of the proposed design more prominently to distinguish it from prior work.
The article presents a correctly described research method. The authors clearly set the goal, prepared the sensor design, subjected it to simulations, optimization, then produced the test object/prototype and subjected it to tests. Obtained results they subjected to appropriate analysis. The conclusions drawn are basically correct. Each conclusion is supported by specific results, such as the influence of different parameters on resonance frequency and voltage, along with quantified improvements (e.g., resonance frequency increased by 4.14%). The inclusion of error percentages (1.43% and 0.14%) when comparing predicted and simulated values demonstrates the robustness of the optimization method.
What I find most missing in the article is a description of real, concrete applications of this sensor, whose parameters have been improved by a few percent. Why would this few percent improvement be decisive in a real-world application. Please provide examples.
Author Response
Dear Reviewer,
Thank you for your detailed review of our work and valuable suggestions. The following is our specific response:
- In my opinion summarizing similar studies together could reduce redundancy and enhance readability.
Answer: Thank you for the reviewer 's advice. I agree with your point of view and reorganize the content of the introduction part, and summarize similar research together. The author has made the following responses and modifications. Literatures [ 8-10 ] and [ 17-20 ] are simplified and marked in red font. It is hoped that the above revisions can further improve the readability of readers.
- While the introduction underscores the technical advancements, it could elaborate more on the broader impacts of the proposed sensor design, such as its potential implications for cost-efficiency or industrial adoption.
Answer: Thank you for the reviewer 's suggestion, and give the corresponding method. This is what you pointed out. We think this is a very important supplementary point. We added the following content in the fourth paragraph of the introduction and marked it in red font.
“The optimized sensor design will reduce the demand for complex structures while improving performance, which has a significant advantage in production and maintenance costs”.
- Authors should emphasize the unique aspects of the proposed design more prominently to distinguish it from prior work.
Answer: Thank you for the reviewer 's advice. In the resubmission, we made a change in the last paragraph of the introduction, hoping to further highlight the uniqueness of this method.
- What I find most missing in the article is a description of real, concrete applications of this sensor, whose parameters have been improved by a few percent. Why would this few percent improvement be decisive in a real-world application. Please provide examples.
Answer: When the preliminary design of the sensor is completed, it is necessary to optimize the structural parameters. The optimized sensor design reduces the demand for complex structures and improves performance. Although the percentage point of parameter improvement seems to be limited in value, it has a significant impact on practical application. For example, the optimized resonant frequency is increased by 4.14 %, which increases the working frequency of the sensor and can more accurately respond to the vibration signal of a specific frequency band. The voltage sensitivity is increased by 9.11 %, which means that the sensor 's ability to monitor the signal is enhanced, which is of great significance for the early detection of geological anomalies in the mining area.
We carried out calibration experiments in the laboratory environment and simulated the mine environment for microseismic signal testing. The results show that the optimized sensor has significant sensitivity and accuracy advantages in the operating frequency range. These experimental results indirectly verify the applicability and potential of the sensor in the actual scene. In order to further verify the practical application effect of the optimized design, we plan to cooperate with relevant industries in the follow-up work to try to apply the sensor to the actual scene to obtain more application data and feedback.
Thank you again for your attention and valuable suggestions for our research, which is of great help to us to improve the quality of the paper and express the significance of the research.
Kind regards,
Jikun Dai